# Separating Representation from Reconstruction Enables Scalable Text Encoders

**Megi Dervishi** [1 2 *]   **Mathurin Videau** [1 *]   **Yann LeCun** [3]

## Abstract

While decoders have rapidly scaled, encoders have remained largely unchanged since BERT. We revisit this disparity by frozen backbone evaluation via probing. Under this lens, the representations of BERT encoders become increasingly *unexploitable* by frozen probes, despite improved perplexity. The misalignment originates in BERT's flat design, which couples representation learning to the token reconstruction loss. We propose **CrossBERT**, a two-part architecture that separates the learning of high-quality encoded representations from the rigid grounding of token reconstruction. This design further enables high masking ratios ($\geq 50\%$) and gradient collection over all tokens via a *Complementary Masking Strategy*, respectively increasing throughput by 1.5 to $2\times$ and sample efficiency by $2\times$. Overall, CrossBERT demonstrates monotonic scaling and superior performance on MTEB(eng, v2) and frozen GLUE benchmarks.

## 1. Introduction

Encoders are critical for a variety of modern applications, ranging from large-scale data curation and retrieval-augmented generation to recommendation systems. However, encoder architectures have remained largely unchallenged since BERT, most improvements stem from scaling training datasets. The research community has focused on elaborate post-training pipelines utilizing pre-trained models merely as initialization (Wang et al., 2022). This heavy reliance on downstream finetuning does not show the shortcomings of the pre-trained backbone, hindering its development. Dervishi et al. (2025) recently demonstrated the cost of neglecting pre-training: frontier pre-trained backbones

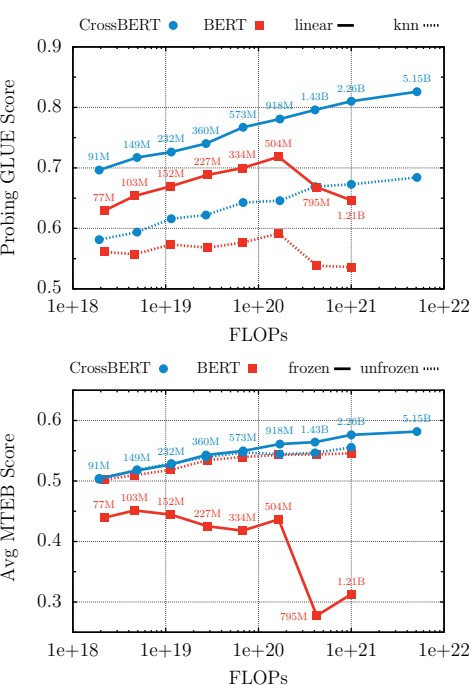

*Figure 1.* **Top.** Frozen evaluation of encoders on GLUE, linear and KNN probes are fitted on the average representation of a frozen backbone. **Bottom.** MTEB(eng, v2) score. '(frozen)' means only the pooler is finetuned on top of the frozen features of the encoder i.e frozen backbone. '(unfrozen)' means that the full network (including the backbone) is finetuned end-to-end. Both are finetuned only on MS-MARCO for one epoch with hard-negatives.

like ModernBERT (Warner et al., 2025) and NeoBERT (Breton et al., 2025) are vastly overtrained relative to their size.

Concurrently, the vision encoder community has continued to improve pre-trained backbones via scaling strategies (Bolya et al., 2025; Sun et al., 2023; Oquab et al., 2023; Siméoni et al., 2025) and novel training recipes (Fu et al., 2024; Chen et al., 2020; Caron et al., 2021; Darcet et al., 2025; Assran et al., 2023; Bardes et al., 2021; Garrido et al., 2024; He et al., 2022). Since these encoders are often utilized frozen, i.e. without modifications, evaluation standards have naturally prioritized frozen benchmarks. Therefore encouraging pre-training innovations. Effective representation learning is defined by the interpretability and versatility of frozen embeddings.

*Equal contribution   [1]FAIR Meta [2]Paris Dauphine University [3]New York University. Correspondence to: Megi Dervishi <megi.dervishi@meta.com>, Mathurin Videau <mvideau@meta.com>.

*Proceedings of the $43^{rd}$ International Conference on Machine Learning*, Seoul, South Korea. PMLR 306, 2026. Copyright 2026 by the author(s).

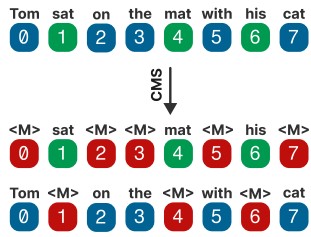
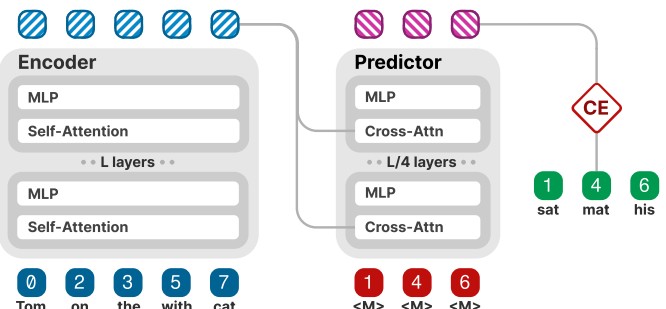

*Figure 2.* **Left.** The Complementary Masking Strategy (CMS) augments a batch of tokens into two complementary masked views by replacing tokens with `<MASK>`. Masked tokens are in red; the unmasked tokens of the two views are in green and blue. Numbers indicate positional indices. **Right.** CrossBERT predicts one view (green) from the other (blue), and vice-versa. Both views are processed in parallel with an attention mask isolating one view from the other, preventing information leakage. The encoder takes the unmasked tokens and their positional indices as input. The lightweight predictor takes `<MASK>` placeholders and their positional indices; it never sees the masked tokens' content. The predictor attends to the encoder's output representations via cross-attention and is trained to reconstruct the complementary view with a Cross-Entropy (CE) loss. Solid color boxes denote tokens (red `<MASK>`, blue/green true values); hatched boxes denote representations (blue from the encoder, pink from the predictor).

Motivated by recent architectural insights in the vision community, we revisit the design and evaluation of text encoders. Specifically, we conduct a comprehensive scaling analysis, evaluating the representations of frozen pretrained backbones alongside standard finetuning protocols. This shift in perspective reveals a counterintuitive phenomenon: as modern BERT models scale, their features become increasingly unexploitable by frozen probes. To resolve this bottleneck, we introduce CrossBERT. Inspired by He et al. (2022); Fu et al. (2024), CrossBERT decouples representation learning from token reconstruction by appending a lightweight cross-attention predictor to the final backbone layers. This architectural shift ensures that the backbone focuses entirely on feature extraction, while token reconstruction is isolated within the predictor. We demonstrate that this novel design preserves representation quality as the model scales. Furthermore, even when evaluated in a data-constrained contrastive setup using only MS-MARCO, CrossBERT mitigates saturation and exhibits a significantly superior scaling trends compared to standard baselines.

Throughout the paper, our baseline BERT is a modern implementation (see Section 5.1). BERT and CrossBERT differ only on the architecture structure: flat vs bipartite. In Section 2 we explain our conjecture for the failure mode of the BERT architecture and how CrossBERT solves it. In Section 3 we detail the architecture of CrossBERT and present its advantages compared to the usual BERT models. In Section 4 we summarize the new frozen evaluation methods that we use to measure solely the pre-training performance. Finally we present our experimental results and discussion in Section 5 and conclude with future work in Section 8.

**Contributions**

**C1. The CrossBERT Architecture.** We introduce a bipartite encoder that ensures consistent performance scal-

ing on frozen evaluations. To the best of our knowledge, this is the first Masked Autoencoder for text.

**C2. High-Efficiency Training via High Masking Ratio.** We demonstrate CrossBERT's robustness at masking ratios $> 50\%$, accelerating training throughput by $\approx 1.5$–$2\times$. Furthermore, this tolerance enables a *Complementary Masking Strategy* (CMS), which processes the inverse mask in parallel. Effectively doubling the sample efficiency by collecting gradients from every token in the sequence.

**C3. Scaling Laws under Frozen Evaluation.** We conduct a scaling analysis ($2 \times 10^{18}$ to $1 \times 10^{21}$ FLOPs) to quantify intrinsic representation quality. This exposes a performance gap in standard BERTs and validates CrossBERT's superior extractability, an advantage that we show extends to MTEB tasks.

## 2. CrossBERT: Intuition

**Problem.** BERT representations worsen as we scale compute as can be seen in Fig. 1.

**Explanation.** We conjecture that the reason behind such degradation stems from the Masked Language Modeling (MLM) objective (Devlin et al., 2019) applied on the flat design of the BERT architecture.

MLM trains an encoder to reconstruct a corrupted input sequence. Given a sequence of tokens $X = \{x_1, \ldots, x_N\}$, a subset of indices $\mathcal{M}$ is selected for masking. The tokens at these positions are replaced by a special token `<MASK>`, yielding a corrupted sequence $\tilde{X}$. The model processes $\tilde{X}$ to generate contextualized representations, and the objective is to minimize the negative log-likelihood of the original tokens $x_m$ at the masked positions:

$$\mathcal{L}_{\text{MLM}} = - \sum_{m \in \mathcal{M}} \log P(x_m \mid \tilde{X}) \qquad (1)$$

where $P(x_m \mid \tilde{X})$ denotes the probability assigned to the true token $x_m$ by the prediction head. *Hence the objective only measures the token reconstruction ability of the model but not the actual quality of its representations.*

Since the "flat" design of the BERT architecture does not explicitly separate representation creation from token reconstruction, the representations remain overly "grounded" in the local signal required to predict missing tokens instead of being versatile high-level abstractions. Some evidence of this phenomenon is displayed in Table 1, where the BERT encoder is probed at different depths. Notably, retrieving representations from earlier layers, rather than the final output, yields slightly improved performance, showcasing the over-specialization of these last layers.

**Solution.** Inspired by Masked Auto-Encoders (MAE) approaches in vision (He et al., 2022; Fu et al., 2024), we propose CrossBERT: a bipartite architecture that separates the heavy lifting of representation creation (Encoder) from the specific task of token reconstruction (Predictor), as illustrated in Fig. 2.

## 3. CrossBERT: Architecture & Advantages

### 3.1. Architecture

A sketch of the architecture is shown in the right panel of Figure 2. The input sequence is partitioned into a visible set (processed by the encoder) and a masked set (processed by the predictor). To preserve sequence order, we encode the position of each token through RoPE, ensuring the encoder and predictor are aware of which positions are missing. The predictor is a few transformer blocks that *only* cross-attend to the encoder representations. By removing self-attention between masked tokens, we force the predictor to act strictly as a "readout" interface that must satisfy its objective solely by querying the encoder's embeddings. Additionally we implement modern architectural optimizations (Warner et al., 2025; Breton et al., 2025) such as RMSNorm.

To set the size of the encoder and predictor we align with MAE (He et al., 2022). The predictor shares the encoder's hidden dimension, but is significantly shallower (approximately one-forth of the encoder's depth). Ablation on the predictor's shape and size can be found in Appendix A. We gain two insights from this ablation. First, the specific aspect ratio (width vs. depth) of the predictor is not significant. Second, while increasing the predictor's capacity can yield some performance gains (with diminishing returns), it comes at the cost of slower training. Since our objective is

*Table 1.* Layer-wise Frozen GLUE Score Analysis on BERT. The scores are obtained by fitting a linear probe on the frozen features as explained in Section 4.1

| Model | last | last-1 | last-2 | last-3 | 20th | Avg. Improv. |
|---|---|---|---|---|---|---|
| BERT 239M | 67.4 | 68.5 | 69.5 | 70.2 | 68.7 | +1.8 |
| BERT 1.21B | 64.6 | 65.8 | 65.0 | 65.9 | 66.6 | +1.2 |

to obtain rich frozen features, we decide to allocate most of the compute to the encoder.

### 3.2. Emerging Advantages

This design choice leads to several advantages: predictor's transfer learning; robustness to higher masking ratios; the ability to use the Complementary Masking Strategy (CMS); better data efficiency and lower training costs.

**Transferability.** The predictor learns to extract information only from the encoder representation during pre-training. Hence, the predictor functions as a learned pooling mechanism. It can be effectively re-used as a warm-started module for downstream finetuning, serving as an efficient bridge between the frozen encoder features and the target task.

**Robustness to higher masking ratios.** BERT architectures typically suffer performance degradation when masking ratios exceed 20–40% (Wettig et al., 2023) (see Appendix C). We challenge this limitation by profiling the frozen representation quality of CrossBERT across multiple masking ratios. As shown in Table 2, CrossBERT exhibits remarkable stability: increasing masking from 20% to 50% incurs a negligible drop in GLUE performance ($-0.7\%$). This resilience confirms that our bipartite architecture successfully insulates representation learning from the difficulty of the reconstruction task. Increasing to higher masking ratios directly translates into reduced training costs, which opens the door to a *Complementary Masking Strategy (CMS)*.

*Table 2.* Average Frozen GLUE score of CrossBERT across different masking ratios keeping the same data budget.

| Masking ratio (%) | 20 | 40 | 50 | 65 |
|---|---|---|---|---|
| **CrossBERT** | 73.8 | 73.7 | 73.1 | 72.4 |

**CMS.** Complementary Masking Strategy is depicted in the left panel of Figure 2. Every batch is augmented with its inverse mask, i.e. creating two complementary views of the original token sequence. This allows the model to predict and learn from every token in the sequence. The model ensures that information from the original sequence does not "leak" into the complementary view within the same batch by applying a unique attention mask on each

view. Hence, both views can be processed simultaneously. For BERT, this approach does not mix well with the 20% – 40 % masking ratio requirement as the inverse view would land on the 60% – 80% ratio. Because CrossBERT remains effective beyond 50% masking, it can learn from both the original and the complementary sequence efficiently.

**Sample efficiency.**  MLM is known for being sample-inefficient; the model learns only from a fraction of tokens per pass since gradients are only computed on masked tokens. However by utilizing CMS, CrossBERT processes the visible tokens and its inverse, effectively seeing all the tokens in the sequence and predicting all the tokens from the predictor in one go. Empirically, our results (Appendix B) confirm that CMS does not damage the performance of the baseline compared to standard masking. Instead, CMS effectively doubles sample efficiency, requiring half the training data to reach the same performances.

**Computational efficiency.**  Even if BERT were to employ CMS, the computational cost would be prohibitively expensive: two forward-backward passes on the full sequence. On the other hand, the encoder of CrossBERT drops masked tokens. Hence the cost of training with CMS is just one forward-backward pass of the encoder and predictor on the full sequence.

## 4. Method

In this section we aim to describe how we prove that CrossBERT works better than the current BERT recipe.

Historically, encoder evaluations have relied heavily on full finetuning for downstream tasks ranging from classification to information retrieval. Doing so, makes it hard to judge whether the final downstream performance is a result of finetuning or original pretrained representation quality. Therefore, we evaluate performance by freezing the encoder, which allows us to directly measure the pretrained representation quality.

We focus on two evaluation benchmarks. While GLUE (Wang et al., 2018) served as the gold standard for early pre-training, the field, especially within the contrastive fine-tuning landscape, has since adopted the MTEB benchmark (Muennighoff et al., 2023) to better assess embedding quality. We now describe how to adapt these established benchmarks to a frozen evaluation protocol.

### 4.1. GLUE frozen evaluation

Given the historical importance of classification in encoder evaluation, we use classification tasks to probe feature quality throughout the training process. Specifically, the output of the last layer is averaged across all tokens to generate a single representation of the sequence. A classifier is then optimized on these frozen features. To ensure efficiency, these probes are fitted in parallel, necessitating only a single forward pass over the training set. Two distinct types of classifiers are used:

**Linear probing.**  The linear probes are optimized via Ridge Regression, which minimizes the standard least-squares error augmented by a $L_2$ penalty term. The loss function is defined as:

$$\mathcal{L}_{\text{Ridge}}(\mathbf{W}) = \|\mathbf{Y} - \mathbf{XW}\|_F^2 + \lambda\|\mathbf{W}\|_F^2 \qquad (2)$$

where $\mathbf{X}$ represents the frozen encoder representations, $\mathbf{Y}$ the target labels, and $\lambda$ the regularization coefficient controlling the penalty strength. The optimal weights $\mathbf{W}^*$ are computed directly via the closed-form analytical solution:

$$\mathbf{W}^* = (\mathbf{X}^\top\mathbf{X} + \lambda\mathbf{I})^{-1}\mathbf{X}^\top\mathbf{Y} \qquad (3)$$

This approach allows for rapid, deterministic fitting across multiple regularization strengths without the need for iterative optimization. Moreover, this procedure is done on GPU leveraging `torch.linalg.solve` making it even faster. In practice, we sweep multiple logspaced $\lambda$ as we notice that some dataset, and especially small ones, are very sensitive to this hyperparameter.

**kNN probing.**  As a non-parametric complement to linear evaluation, k-Nearest Neighbors (kNN) assess the intrinsic geometry of the representation space. For a given query $z$, the prediction is determined by a majority vote among the set of its $k$ closest neighbors $\mathcal{N}_k(z)$, identified by minimizing the distance metric $d$ (either $L_2$ or Cosine):

$$\mathcal{N}_k(z) = \operatorname*{arg\,min}_{\mathcal{S}\subset\mathcal{D},|\mathcal{S}|=k} \sum_{x_j\in\mathcal{S}} d(z, x_j)$$
$$\hat{y} = \operatorname*{arg\,max}_{c\in\mathcal{C}} \sum_{x_i\in\mathcal{N}_k(z)} \mathbb{1}(y_i = c) \qquad (4)$$

where $\mathcal{D}$ is the set of all evaluation points and $\mathcal{C}$ the set of all classes. The implementation runs on GPU and is heavily based on the released code of CAPI (Darcet et al., 2025).

In practice, these evaluations are computationally negligible. For a 250M parameter model, the entire linear and kNN probing process on GLUE takes less than 5 minutes on a single H100. The cost is dominated by the forward pass over the dataset.

### 4.2. MTEB frozen evaluation

We evaluate the richness and adaptability of our frozen representation using the Massive Text Embedding Benchmark (Muennighoff et al., 2023), which spans seven distinct task downstream categories: Classification, Clustering, Pair Classification, Semantic Textual Similarity (STS), Reranking, Retrieval, and Summarization.

*Table 3.* Results on GLUE and MTEB(eng, v1) under both full-finetuning (unfrozen backbone) and frozen probing (Sections 4.1 and 4.2). We report score for MTEB(eng, v1) to be able to compare with previous work. See Appendix E for the scores on MTEB(eng,v2).

| | GLUE | | | | | | | | | | | |
|---|---|---|---|---|---|---|---|---|---|---|---|---|
| Model | Params | FLOPS | tps | MNLI | QNLI | QQP | RTE | SST | MRPC | CoLA | STS | Avg. |
| BERT | 239M | 6.8e19 | 123k | 86.5 | 90.0 | 88.5 | 85.3 | 95.1 | 91.9 | 63.0 | 91.2 | 86.4 |
| CrossBERT | 279M | 4.1e19 | 207k | 86.4 | 90.3 | 88.4 | 85.1 | 94.8 | 91.4 | 65.5 | 91.5 | 86.6 |
| Electra | 258M | 7.5e19 | 116k | 88.7 | 93.3 | 89.1 | 83.4 | 94.6 | 92.0 | 71.0 | 90.0 | 87.8 |
| *ModernBERT* | 352M | 4.9e21 | - | 90.8 | 95.2 | 92.7 | 92.1 | 97.1 | 91.7 | 71.4 | 92.8 | 90.5 |
| *NeoBERT* | 198M | 2.9e21 | - | 88.9 | 93.9 | 90.7 | 91.0 | 95.8 | 93.4 | 64.8 | 92.1 | 88.8 |
| *OptiBERT* | 239M | 7.0e19 | - | 86.6 | 92.1 | 90.3 | 83.2 | 92.6 | 91.0 | 59.6 | 90.8 | 85.8 |
| *DeBERTaV3* | 304M | >2e20 | - | 91.9 | 96.0 | 93.0 | 92.7 | 96.9 | 91.9 | 75.3 | 93.0 | **91.4** |

| | GLUE Linear probe | | | | | | | | | | | |
|---|---|---|---|---|---|---|---|---|---|---|---|---|
| BERT | 239M | 6.8e19 | 123k | 58.6 | 74.8 | 77.7 | 59.9 | 84.9 | 76.7 | 27.4 | 79.5 | 67.4 |
| CrossBERT | 279M | 4.1e19 | 207k | 61.6 | 81.8 | 81.4 | 64.3 | 90.6 | 77.9 | 47.9 | 84.8 | 73.8 |
| Electra | 258M | 7.5e19 | 116k | 69.2 | 83.6 | 82.7 | 66.8 | 86.9 | 79.6 | 58.8 | 87.5 | 76.9 |
| *ModernBERT* | 352M | 4.9e21 | - | 63.7 | 80.0 | 80.7 | 61.0 | 86.7 | 74.0 | 40.7 | 83.0 | 71.2 |
| *NeoBERT* | 198M | 2.9e21 | - | 47.9 | 69.8 | 72.9 | 56.0 | 78.0 | 69.1 | 10.1 | 66.1 | 58.7 |
| *DeBERTaV3* | 304M | >2e20 | - | 76.6 | 86.9 | 85.8 | 75.8 | 86.5 | 81.9 | 67.1 | 90.1 | **81.3** |

| | MTEB(eng, v1) Full Contrastive Finetuning on MS-MARCO | | | | | | | | | | | |
|---|---|---|---|---|---|---|---|---|---|---|---|---|
| | | | | Class. | Clust. | PairClass. | Rerank. | Retriev. | STS | Summ. | Avg. | Overall |
| BERT | 239M | 6.8e19 | 123k | 62.7 | 35.1 | 80.7 | 51.5 | 43.0 | 75.8 | 29.8 | 54.1 | 53.9 |
| CrossBERT | 279M | 4.1e19 | 207k | 67.6 | 31.5 | 80.5 | 52.2 | 42.5 | 75.3 | 31.6 | **54.5** | **54.1** |
| Electra | 258M | 7.5e19 | 116k | 62.1 | 27.7 | 77.9 | 48.8 | 34.3 | 72.4 | 29.1 | 50.3 | 49.0 |
| *ModernBERT** | 352M | 4.9e21 | - | 62.4 | 38.7 | 65.5 | 50.1 | 23.1 | 68.3 | 27.8 | 46.9 | - |
| *NeoBERT** | 198M | 2.9e21 | - | 61.6 | 40.8 | 76.2 | 51.2 | 31.6 | 74.8 | 30.7 | 51.3 | - |
| *OptiBERT*× | 239M | 7.0e19 | - | 67.5 | 36.1 | 75.8 | 48.1 | 23.3 | 79.7 | 30.0 | 51.5 | - |
| *DeBERTaV3** | 304M | >2e20 | - | 45.9 | 16.4 | 45.0 | 40.8 | 4.0 | 40.1 | 29.9 | 27.1 | - |

* Full finetuning on much larger dataset including MSMARCO, StackOverFlowDupQuestion, Fever, STS12, and STSBenchmark and AllNLI.

× Full finetuning on AllNLI only.

| | MTEB(eng, v1) Contrastive Finetuning over frozen features on MS-MARCO | | | | | | | | | | | |
|---|---|---|---|---|---|---|---|---|---|---|---|---|
| BERT | 239M | 6.8e19 | 123k | 60.1 | 29.7 | 64.6 | 43.7 | 19.2 | 62.7 | 30.5 | 44.3 | 42.1 |
| CrossBERT | 279M | 4.1e19 | 207k | 65.7 | 37.1 | 78.0 | 51.4 | 40.7 | 72.3 | 30.7 | **53.7** | **53.6** |
| Electra | 258M | 7.5e19 | 116k | 41.3 | 12.8 | 35.4 | 32.4 | 0.0 | 32.8 | 26.9 | 26.2 | 22.4 |

**Contrastive probing.** The above downstream tasks require sentence/document representations and a coherent, well-structured representation space, which is not tackled by the pre-training objective. Contrastive finetuning aims to address these gaps. While existing literature typically performs this via contrastive finetuning on an unfrozen backbone, we explore two configurations: a standard unfrozen backbone and a frozen-backbone approach where only an attention based pooler is optimized on top of fixed features.

The token-level encoded representations are pooled into a single sentence/document representation via a learnable lightweight adapter, consisting of a few Transformer blocks with cross-attention. Let the pooled representations of a passage (resp. query) be $p$ (resp. $q$). Given a query $q$, the contrastive loss, aims to pull closer a related passage $p$ and the query $q$ in representation space while pushing away semantically similar unrelated passages $p^-$ (hard-negatives). The contrastive loss, as defined in Chen et al. (2020), is

$$\mathcal{L}_c = -\frac{1}{N} \sum_{i=1}^{N} \log \frac{e^{\phi(q_i, p_i)}}{e^{\phi(q_i, p_i)} + \sum_{n \in \mathcal{S}_i} e^{\phi(q_i, p_{in}^-)}} \quad (5)$$

where $\phi(q_i, p_i)$ is the cosine similarity between a query $q_i$

and a passage $p_i$, and $p_{in}^-$ denotes the hard negatives for each query.

Since CrossBERT already features an adaptable component (the predictor), we re-purpose it to warm-start the adapter for the contrastive learning phase, while keeping the encoder frozen. As detailed in our ablation study (Appendix D), this strategy accelerates convergence and improves MTEB scores by $\approx 2\%$.

### 4.3. Evaluations at different scales

Model performance and optimal hyperparameters follow predictable power-law trends relative to the total compute budget $C$ (Kaplan et al., 2020; Bi et al., 2024). We define $C$ as:

$$C = F_N D \quad (6)$$

where $D$ is the number of pre-training tokens and $F_N$ are the FLOPs per token for a forward-backward pass. For a standard BERT transformer with sequence length $S$, layers $L$, and non-embedding parameters $N$ (Dervishi et al., 2025) we have:

$$F_N = 6N + 12dLS \quad (7)$$

However, CrossBERT modifies $F_N$ based on the masking ratio $r$, linearly interpolating between the encoder ($F_N^{\text{enc}}$) and the cross-attention predictor ($F_N^{\text{pred}}$):

$$F_N = F_N^{\text{enc}}(1 - r) + F_N^{\text{pred}}r \qquad (8)$$

This demonstrates that as the masking ratio increases, the total computational cost linearly interpolates between the encoder and the predictor. To scale up we need to increase model sizes $F_N$ and dataset sizes $D$ in tandem.

## 5. Experiments

### 5.1. Setup

**Models.** We train three models that share the same corpus, tokenizer, and encoder backbone: BERT, Electra, and CrossBERT. BERT and Electra are flat, whereas CrossBERT is bipartite (Section 3). BERT and CrossBERT use the MLM objective of Section 2, while Electra replaces it with replaced-token detection (RTD) (Clark et al., 2020): a small auxiliary generator substitutes a fraction of the input tokens with plausible alternatives, and the encoder predicts at each position whether the token is original or replaced. For context, we also report the published results of ModernBERT (Warner et al., 2025), NeoBERT (Breton et al., 2025), DeBERTaV3 (He et al., 2023), and OptiBERT (Dervishi et al., 2025), all of which are trained with substantially more data and compute.

**Data.** We choose a subset of DCLM (Li et al., 2024) as our training corpus ($\approx$ 4T tokens). Across all experiments, the masking ratio is set to 20% for BERT and 50% for CrossBERT. Additionally, all CrossBERT models leverage CMS (see Section 3.2). All models rely on the RoBERTa tokenizer (Liu et al., 2019).

**Codebase.** Our codebase is based on PyTorch (Paszke et al., 2019) and Lingua (Videau et al., 2024), opting for Fully Sharded Data Parallelism (FSDP) and `torch.compile` for maximum throughput and reduced memory footprint. *Handling Dynamic Shapes:* Compilation requires static graphs, which conflicts with random MLM masking. We resolve this by enforcing a fixed count of masked tokens per GPU (Per GPU Batch size × Mask Ratio) and permuting each mask in the sequence via random permutations (`torch.randperm`). This ensures static tensor shapes without compromising masking randomness.

**Hyperparameters.** Following Bi et al. (2024), we fit power laws for both Batch Size (BSZ) in total tokens and Learning Rate (LR) sweeping model sizes ranging from (50M to 700M) with a LR (resp. BSZ) logspaced from $10^{-4}$ to $10^{-2}$ (resp. $10^4$ to $5 \times 10^6$). After keeping only the top 3

performing models for the fitting, we obtain the following results for CrossBERT:

$$\text{BSZ}_{\text{CrossBERT}}(C) = 10^4 \times C^{0.092}$$
$$\text{LR}_{\text{CrossBERT}}(C) = 564.6 \times C^{-0.279}$$

For the BERT setup, we simply reuse the hyperparameters found by (Dervishi et al., 2025)

$$\text{BSZ}_{\text{BERT}}(C) = 17.38 \times C^{0.24}$$
$$\text{LR}_{\text{BERT}}(C) = 69.18 \times C^{-0.24}$$

Appendix H shows the hyperparameter heatmap for the CrossBERT sweeping.

**Eval on GLUE.** For Linear Probing, we fit linear heads on frozen features along logarithmic data regimes ($\{1, \ldots, 10^4\}$ samples) and L2-regularization strengths ($\lambda \in [1, 10^4]$). For each task we report the best score across all $\lambda$ for the largest available sample size. For kNN Probing, non-parametric evaluation with kNN are conducted using diverse configurations by sweeping across distance metrics ($L_2$, Cosine) and neighborhood sizes ($k \in \{1, 3, 10, 30\}$). We report the maximum score across all $k$ and distances.

**Eval on MTEB.** We use the MS-MARCO training set (500k queries) with hard negatives. No instruction templates are applied. We finetune contrastively for one epoch, with a batch size of 512 and a learning rate of $5 \times 10^{-5}$ (cosine decay to $5 \times 10^{-7}$). To be comparable with current published models, we use both an unfrozen and frozen backbone.

**Single-scale runs.** All three models share the same encoder backbone (28 layers, hidden dimension 768); CrossBERT additionally couples this encoder to a 6-layer cross-attention predictor. The models are trained on 50B tokens with a data-to-model ratio of $35 : 1$ that deliberately exceeds the compute-optimal $\approx 15 : 1$ to avoid undertraining. The learning rate is fixed at $6 \times 10^{-4}$ and the global batch size at $\approx 393$k.

**Scaling sweep.** Dervishi et al. 2025 showed that the data-to-model ratio $F_N/D$ heavily impacts performance and should be kept around 10:1 to 100:1. In all our scaling experiments we fix the ratio at $20 : 1$ and generate a suite of models spanning a total compute range from $2 \times 10^{18}$ to $1 \times 10^{21}$ FLOPs. To optimize performance at every scale, we determine batch size and learning rate by sweeping values across smaller models (50M to 700M) and extrapolating the optimal settings via a power law fitting). In total we train 8 BERT and 9 CrossBERT models, see Appendix J.

### 5.2. Results: Single-scale Runs

Table 3 reports the single-scale runs of our three models alongside the current literature.

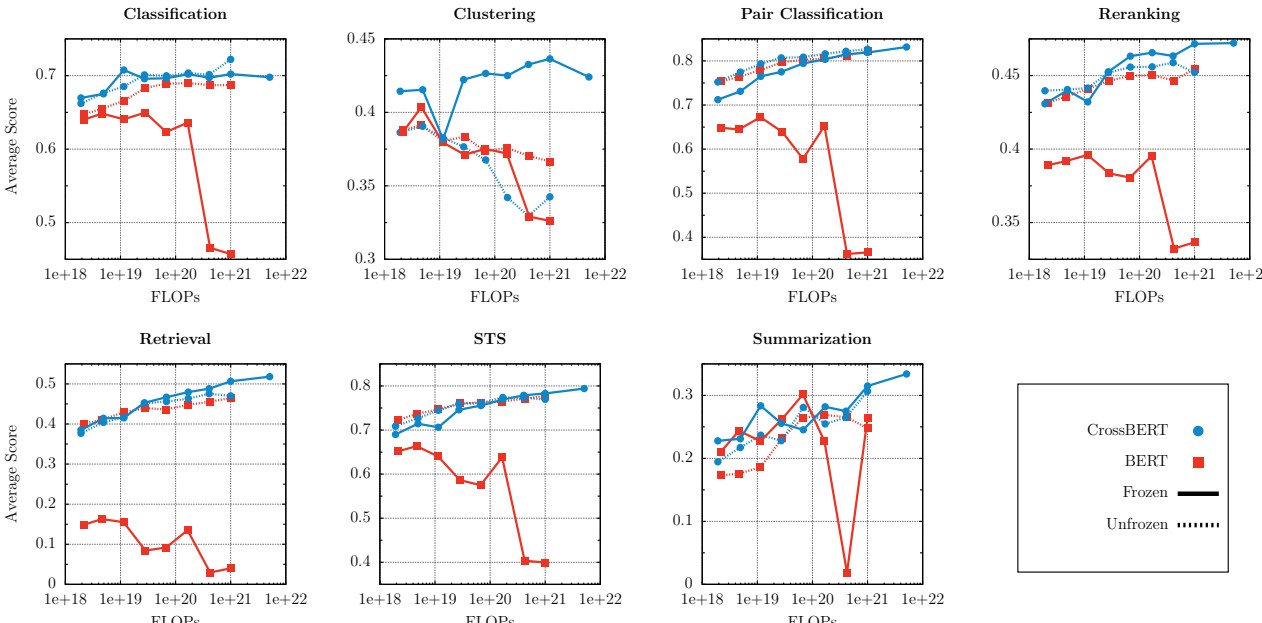

*Figure 3.* Scaling trend of BERT vs CrossBERT after contrastive finetuning under frozen and unfrozen backbone for MTEB(eng, v2). All models are trained for one epoch on MS-MARCO as described in Section 5.1

**Training efficiency and throughput.** CrossBERT demonstrates superior computational efficiency. In terms of training throughput, it achieves 207k tokens/sec compared to the baseline's 123k tokens/sec, representing a $1.68\times$ speedup in wall-clock time on H100 GPUs. Furthermore, when compared to the existing literature, CrossBERT remains highly competitive despite utilizing approximately $100\times$ less total compute than models like ModernBERT and NeoBERT. Against OptiBERT, a model of similar scale, CrossBERT requires $\approx 40\%$ fewer FLOPs while delivering higher performance on both GLUE and MTEB(eng, v1).

**Full-finetuning performances.** Despite this massive reduction in compute, CrossBERT does not compromise on quality. On the GLUE benchmark, it matches our robust baseline (86.6 vs. 86.4). More notably, on the MTEB(eng, v1) benchmark (Full Contrastive Finetuning), CrossBERT achieves the highest average score of **54.5**, outperforming both NeoBERT (51.3) and ModernBERT (46.9) which used much more finetuning data.

**Robustness of frozen representations.** The most significant advantage of CrossBERT lies in the versatility of its frozen features. We observe three critical behaviors:

- **Minimal degradation:** When switching from full finetuning to frozen adaptation, the standard BERT baseline suffers a substantial drop of over 10 points on MTEB(eng, v1) ($53.9 \rightarrow 42.1$). In contrast, CrossBERT retains 99% of its performance, scoring

53.6 in the frozen setting. This indicates that the pretrained features are easily adaptable.
- **Retrieval capability:** This robustness is most visible in the Retrieval task, where CrossBERT more than doubles the score of the baseline (40.7 vs. 19.2), proving that the model captures semantic similarity without needing deep task-specific adaptation.
- **Linear separability:** On the GLUE Linear Probe, CrossBERT surpasses the BERT baseline by a large margin (+6.4 points). This confirms that our approach prevents the over-specialization often seen in standard MLMs, producing high-level abstractions that are linearly separable and directly usable for downstream tasks. While a performance gap of $\approx 13$ points remains between the linear probe and full finetuning, this is partly attributable to the simplicity of averaging encoder outputs; utilizing more sophisticated probing mechanisms, such as attention-based pooling, would likely reduce this gap further. Electra scores even higher than CrossBERT on this probe, but this advantage does not carry over to the sentence-embedding tasks examined next.

**MLM versus RTD on a flat backbone.** On token-level probes, Electra outperforms BERT and CrossBERT, reaching 87.8 on GLUE full finetuning and 76.9 on the GLUE linear probe. However, on sentence-level embedding tasks it collapses. The unfrozen backbone benchmarks at 49.0 on MTEB(eng,v1) whereas the frozen one drops to 22.4 with a Retrieval score of 0.0. This collapse mirrors prior find-

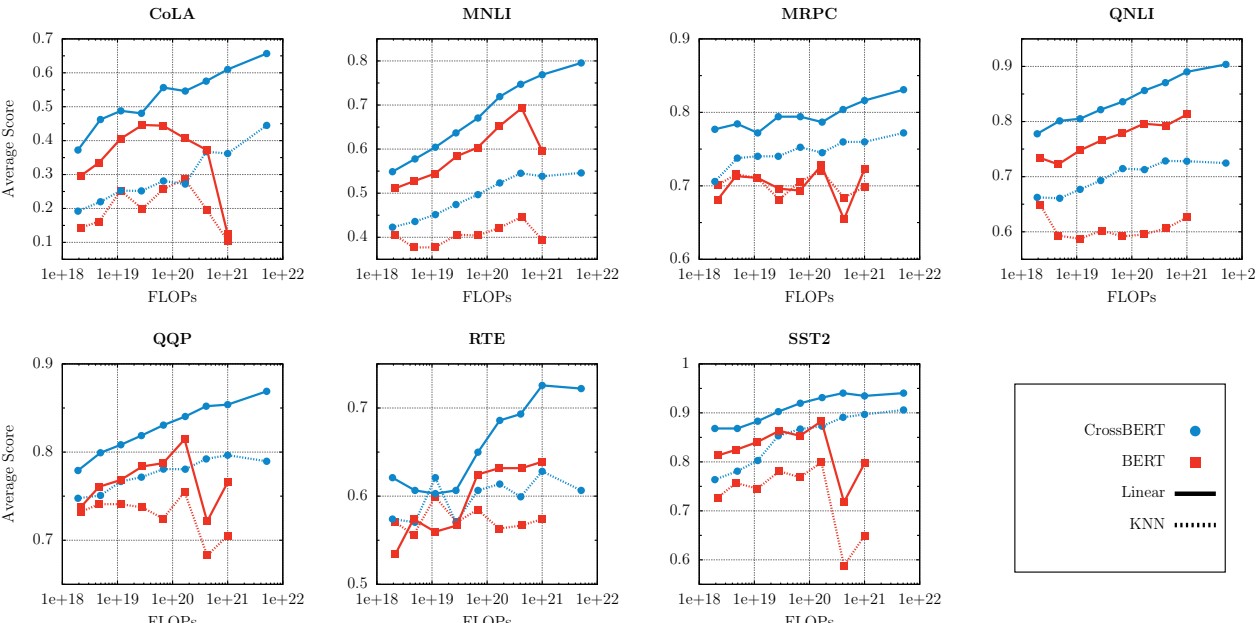

*Figure 4.* Scaling trends of BERT vs CrossBERT using linear and kNN probing for GLUE under frozen backbone(see Section 5.1).

ings that RTD distorts sentence-embedding geometry (Rep et al., 2024; Warner et al., 2025). Two conclusions follow. First, neither swapping the objective on a flat backbone (Electra) nor keeping MLM on a flat backbone (BERT) yields versatile frozen representations; the bipartite separation of representation from reconstruction is what produces them. Second, GLUE-style classification probes alone cannot diagnose representation quality, since MTEB exposes failures that GLUE hides.

### 5.3. Results: Scaling Sweep

Figure 1 aggregates the GLUE and MTEB(eng,v2) scores of BERT and CrossBERT, while Figures 3 and 4 break them down per task. Our sweep reveals four insights into the scaling behavior of masked language models.

**Degradation of frozen BERT representations.** The most striking trend is the sharp divergence in scaling laws between the two models. While standard BERT benefits from scaling when it is unfrozen and fully finetuned, its *frozen* performance suffers a brutal drop as model size increases. Specifically, for the two largest configurations ($> 500M$ parameters), BERT's frozen performance drops below that of the smallest model in the sweep ($10\times$ smaller). Since the data-to-model ratio (20:1) is held above the compute-optimal ratio (15:1), undertraining cannot explain the drop. This confirms a severe misalignment between the standard MLM objective and semantic embedding tasks: as the model scales, it becomes increasingly specialized for token reconstruction at the expense of versatile, high-level abstractions.

**Task-specific sensitivity on GLUE.** This degradation manifests non-uniformly across tasks (Figure 4). On the GLUE benchmark, the sudden decline of performance does not necessarily appear at the same point. For example MNLI accuracy drops only for the last model while performances for the rest seem to drop earlier.

**MTEB(eng, v2) superior scaling and saturation profiles.** In contrast, CrossBERT exhibits robust, monotonic scaling across all metrics. Most notably, CrossBERT fundamentally alters the relationship between pre-training and adaptation:

- **The adaptation gap & equalization on MTEB(eng, v2):** Standard BERT requires full finetuning to bridge a massive performance deficit ($\approx 15$ points) between its frozen and unfrozen states. While this heavy downstream adaptation can eventually equalize performance, i.e. masking pre-training deficiencies by bringing BERT closer to CrossBERT, it comes at a significant compute cost. In contrast, CrossBERT's frozen representations are naturally aligned, sitting at worse around one point below the fully fine-tuned optimum.
- **Structural retrieval capability:** We observe a fundamental distinction in retrieval tasks. Standard MLM effectively flatlines near zero at all scales, indicating a structural inability to learn dense retrieval without supervision. In contrast, CrossBERT builds these capabilities naturally, scaling linearly with compute.
- **Frozen outperforming unfrozen finetuning:** At larger scales, CrossBERT with frozen adaptation begins to outperform even the unfrozen baselines. This

indicates that CrossBERT scales more effectively than standard BERT, avoiding early saturation. It produces representations that are naturally richer, rendering the heavy process of unfrozen finetuning unnecessary, and eventually inferior, to a lightweight adaptation of the frozen features.

**Training stability.** Self-supervised pre-training can suffer from unstable optimization, so we monitor training across the full scaling range. Two observations indicate that CrossBERT trains stably. First, the pre-training validation loss decreases smoothly and tracks the expected scaling law at every compute budget (Appendix F). Second, downstream probing performance rises steadily over the course of training rather than oscillating or collapsing (Figure 6). We observe no divergence across scales. In contrast higher scale BERT trainings suffer from significant instability and loss spikes (see Appendix I).

## 6. Related Works

**Standard Encoders.** Since BERT (Devlin et al., 2019) and RoBERTa (Liu et al., 2019), encoders have relied on *flat* architectures where representation and reconstruction are mixed. Recent updates like ModernBERT (Warner et al., 2025) and NeoBERT (Breton et al., 2025) scale this design but inherit its structural inefficiency. As a result, brute-force scaling of this approach yields diminishing returns for representation quality (Dervishi et al., 2025).

**Contrastive Learning.** To compensate for this misalignment, the field relies on heavy post-training (Gao et al., 2021; Wang et al., 2022; Li et al., 2023) or token-level objectives like MEXMA (Janeiro et al., 2025). These methods effectively treat the symptoms, but we posit that a better aligned pre-training can reduce the amount of adaptation needed to obtain the final model.

**Architecture Design.** In computer vision, Masked Autoencoders (He et al., 2022) established the efficacy of asymmetric designs, where a lightweight decoder reconstructs pixels from highly masked inputs. Fu et al. (2024) further demonstrated that reducing interaction between representation and reconstruction, using cross attention only, enhances produced features. In NLP, while T5 (Raffel et al., 2020) also utilizes a bipartite structure, it relies on a heavy decoder optimized for autoregressive text generation employing both self and cross attention. In contrast, CrossBERT adopts asymmetry strictly for representation learning: rather than generating text, we employ a lightweight, cross attention only, predictor solely to offload the reconstruction burden, ensuring the encoder optimizes for semantic abstraction rather than token prediction.

## 7. Limitations

Our analysis is empirical. We identify the failure mode of flat MLM through controlled experiments (Table 1 and Figure 1), but we do not give a theoretical account of why coupling representation and reconstruction degrades frozen features as models grow. Our conclusions also rest on models no larger than a few billion parameters; every trend we observe is monotonic across this range, yet we cannot formally exclude qualitatively different behavior at the much larger scales typical of decoders. Finally, we find that RTD (Electra) yields strong token-level probes but poor sentence embeddings, and a mechanistic explanation of this collapse lies outside the scope of this work.

## 8. Conclusion & Future Work

We revisited the design of text encoders by shifting the evaluation focus from full finetuning to frozen representation quality. This change in perspective revealed that flat BERT-like architectures trained with MLM suffer from a fundamental misalignment: as compute scales, representations become increasingly unexploitable for downstream tasks, overspecializing on reconstruction at the expense of versatility. CrossBERT resolves this by insulating representation learning from the token prediction task. Beyond improving training efficiency, our results show an interesting trend: at large scales, CrossBERT produces frozen features that outperform fully finetuned baselines. This result challenges the expensive downstream contrastive adaptation of standard encoders, demonstrating that the right pre-training incentives can produce significantly richer representations needing less heavy adaptation to be effective.

Several directions follow naturally from these findings. The bipartite design is agnostic to the reconstruction objective, so MLM could be swapped for alternative pretext tasks better suited to representation learning. A systematic comparison with T5-style pre-training, where the predictor is replaced by an auto-regressive decoder would be an interesting avenue to explore.

## Acknowledgements

We thank Badr Youbi Idrissi and João Maria Janeiro for the helpful discussions and feedback that shaped this work.

## Impact Statement

This paper presents work whose goal is to advance the field of Machine Learning. There are many potential societal consequences of our work, none which we feel must be specifically highlighted here.

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

## A. Predictor shape design ablation

Table 4 analyzes the trade-off between predictor size and downstream performance. At the lower bound, the 8M parameter predictor acts as a representational bottleneck, noticeably impairing model quality. However, increasing capacity to just 25M yields a substantial boost. Beyond this point, we observe diminishing returns; scaling further to 50M or 85M incurs higher computational costs for only marginal performance gains. While we ultimately selected the 42M configuration (one-fourth of the encoder depth, grey row) to maintain dimensional alignment with the backbone, the data suggests that the 25M variant remains a highly competitive alternative, promising faster pretraining speed.

| Predictor | | | GLUE | | | | | | | | |
|---|---|---|---|---|---|---|---|---|---|---|---|
| Params | Dim | Layers | MNLI | QNLI | QQP | RTE | SST2 | MRPC | CoLA | STS | Avg. |
| 8M | 192 | 12 | 58.2 | 78.4 | 80.1 | 64.6 | 87.6 | 72.8 | 45.3 | 82.2 | 71.2 |
| 25M | 384 | 12 | 61.5 | 81.6 | 81.4 | 62.0 | 88.5 | 78.0 | 49.5 | 85.1 | 73.4 |
| 50M | 576 | 12 | 62.8 | 81.7 | 81.5 | 62.1 | 89.2 | 78.9 | 47.8 | 86.6 | 73.8 |
| 85M | 768 | 12 | 63.4 | 81.9 | 81.9 | 63.9 | 91.0 | 76.0 | 48.4 | 86.7 | 74.2 |
| 42M | 768 | 6 | 61.6 | 81.8 | 81.4 | 64.3 | 90.6 | 77.9 | 47.8 | 84.8 | 73.8 |

*Table 4.* Impact of varying the shape and size of the predictor. The encoder has 236.8M parameters (768 dim, 28 layers). All model are trained on same exact setup as Table 3

## B. Complementary Masking ablation

| Model | CMS | MNLI | QNLI | QQP | RTE | SST2 | MRPC | CoLA | STSB | Avg. |
|---|---|---|---|---|---|---|---|---|---|---|
| CrossBERT 279M | ✗ | 61.4 | 81.4 | 81.3 | 62.8 | 88.9 | 74.2 | 48.7 | 85.8 | 73.1 |
| CrossBERT 279M | ✓ | 61.6 | 81.8 | 81.4 | 64.2 | 90.6 | 77.9 | 47.8 | 84.8 | **73.8** |

*Table 5.* Effect of CMS on GLUE Linear probe. All results are obtained under the same settings of Table 3

## C. BERT Masking ablation

| BERT mask | CoLA | MNLI | MRPC | QNLI | QQP | RTE | SST2 | STSB | Avg. |
|---|---|---|---|---|---|---|---|---|---|
| 20% | 27.4 | 58.6 | 76.7 | 74.8 | 77.7 | 59.9 | 84.9 | 79.5 | 67.4 |
| 30% | 18.8 | 57.0 | 73.0 | 74.1 | 77.9 | 54.2 | 80.7 | 77.1 | 64.1 |
| 40% | 11.0 | 53.6 | 69.1 | 73.5 | 75.0 | 57.0 | 75.1 | 73.4 | 61.0 |
| 65% | 22.6 | 50.0 | 67.2 | 70.0 | 73.4 | 55.2 | 75.1 | 65.7 | 59.9 |

*Table 6.* GLUE scores for BERT models trained with different masking rates.

## D. Predictor MTEBv2 contrastive finetuning ablation

| Model | Warm-start | Class. | Clust. | PairClass. | Rerank. | Retriev. | STS | Summ. | Avg. | Overall |
|---|---|---|---|---|---|---|---|---|---|---|
| CrossBERT 279M | ✗ | 68.7 | 38.3 | 76.4 | 43.4 | 37.8 | 68.9 | 28.7 | 51.8 | 53.7 |
| CrossBERT 279M | ✓ | 70.5 | 40.8 | 77.9 | 44.1 | 43.9 | 71.7 | 26.7 | **53.7** | **56.7** |

*Table 7.* Contrastive finetuning under frozen backbones on MSMarco for one epochs as described in Section 5.1

## E. MTEB(eng,v2) Results of the Single-scale runs.

| MTEB(eng, v2) Full Contrastive Finetuning on MS-MARCO | | | | | | | | | | | | |
|---|---|---|---|---|---|---|---|---|---|---|---|---|
| | | | | Class. | Clust. | PairClass. | Rerank. | Retriev. | STS | Summ. | Avg. | Overall |
| BERT | 239M | 6.8e19 | 123k | 67.7 | 39.3 | 80.8 | 43.9 | 45.7 | 75.3 | 28.7 | 54.5 | **57.3** |
| CrossBERT | 279M | 4.1e19 | 207k | 72.2 | 35.5 | 80.5 | 44.8 | 44.2 | 74.6 | 32.0 | **54.8** | 57.0 |
| Electra | 258M | 7.5e19 | 116k | 62.1 | 27.7 | 77.9 | 48.8 | 34.4 | 72.4 | 29.1 | 50.3 | 49.0 |
| MTEB(eng, v2) Contrastive Finetuning over frozen features on MS-MARCO | | | | | | | | | | | | |
| BERT | 239M | 6.8e19 | 123k | 64.9 | 37.3 | 64.5 | 39.4 | 20.5 | 61.6 | 31.8 | 45.7 | 45.9 |
| CrossBERT | 279M | 4.1e19 | 207k | 70.4 | 40.8 | 77.9 | 44.1 | 43.9 | 71.7 | 26.7 | **53.6** | **56.7** |
| Electra | 258M | 7.5e19 | 116k | 43.9 | 29.5 | 35.4 | 33.0 | 0.3 | 31.9 | 19.4 | 27.6 | 26.1 |

*Table 8.* MTEB(eng,v2) results of our models under contrastive finetuning over frozen and unfrozen backbone (full-finetuning).

## F. Learning curve of scaling laws

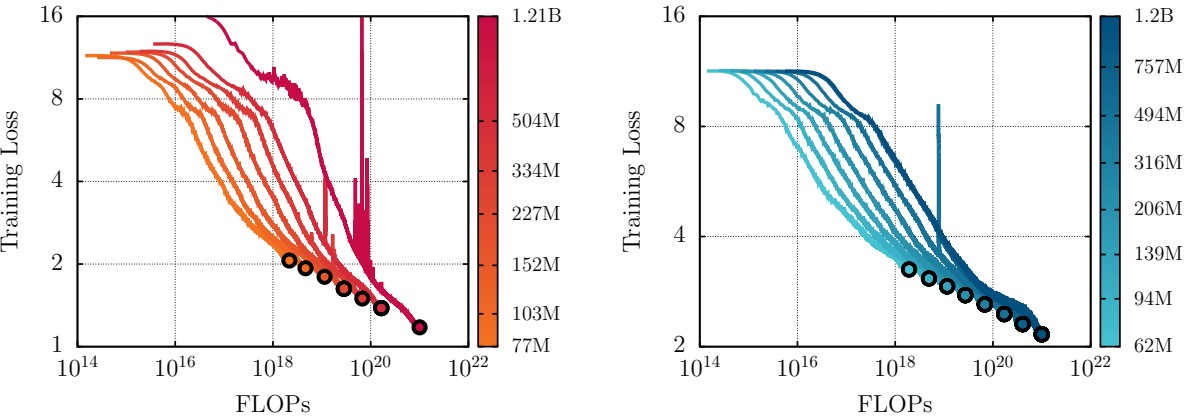

*Figure 5.* Learning curves for different BERT(left) and CrossBERT(right) setup specified in model list Appendix J. Each dot represents the validation loss on wikipedia and dclm(Li et al., 2024).

## G. Downstream performance evolution during training

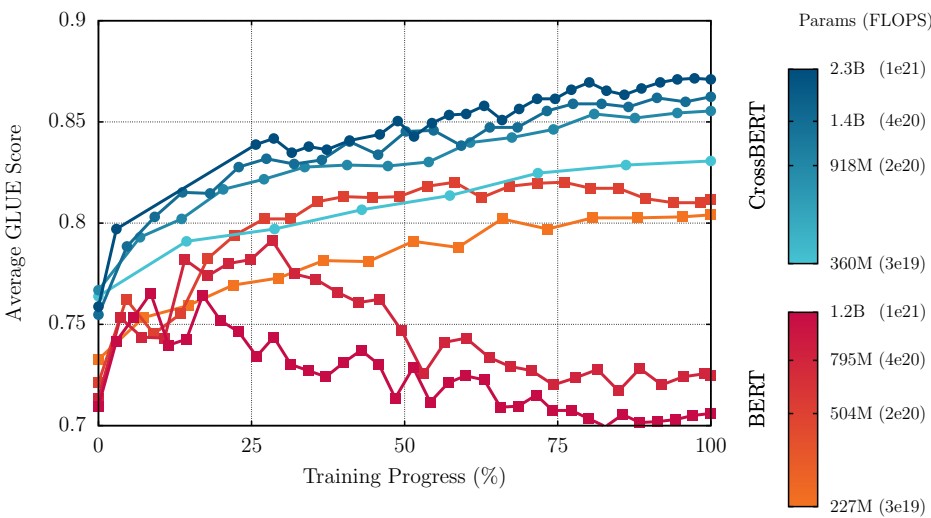

*Figure 6.* Average classification performance across 10 tasks monitored during training for BERT (red, ■) and CrossBERT (blue, ●).

## H. Learning rate and batch size sweeps

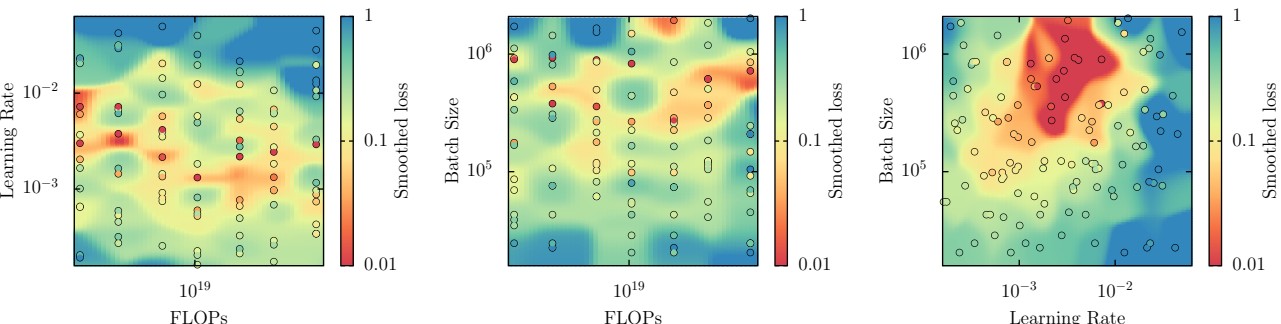

*Figure 7.* CrossBert heatmap for lr and batch size, Left: flops vs lr, Middle: flops vs batch size, Right: lr vs batch size, the size of the point is proportionnal to total compute budget spend for training.

## I. Large scale training stability

Because scaling BERT resulted in a severe degradation of downstream performance, we further investigated this collapse across various training setups. To determine if this instability is specific to BERT, we additionally scaled Electra to 1B parameters. As shown in Table 9, both BERT and Electra exhibit a similar performance collapse at the 1B scale. For BERT, altering standard hyperparameters—including data and model seeds, learning rate, and batch size—failed to prevent the degradation. While specific hyperparameter tuning allowed Electra to avoid collapsing, this extreme sensitivity highlights the inherent brittleness of training standard 'flat' architectures at scale.

Conversely, CrossBERT is highly robust, achieving strong out-of-the-box performance without requiring such exhaustive hyperparameter sweeps. When scaling further to 2B parameters, standard configurations initially failed to converge. However, we found that simply reducing the initialization standard deviation from the conventional 0.02 (standard across ViT and BERT implementations) to 0.015 successfully stabilized the training. Furthermore, while Electra historically demonstrates superior performance at smaller scales, CrossBERT 1B significantly outperforms Electra 1B on GLUE, confirming the superior scaling trajectory of our proposed architecture.

| Model | MNLI | QNLI | QQP | RTE | SST2 | MRPC | COLA | STSB | Avg. |
|---|---|---|---|---|---|---|---|---|---|
| BERT 1B | 60.3 | 78.6 | 73.5 | 60.3 | 71.2 | 70.8 | 19.3 | 70.1 | 63.0 |
| BERT 1B* | 59.6 | 81.3 | 76.6 | 63.9 | 79.7 | 72.3 | 12.3 | 71.6 | 64.6 |
| Electra 1B | 59.0 | 80.7 | 78.6 | 64.6 | 72.5 | 80.4 | 16.7 | 79.8 | 66.5 |
| Electra 1B* | 75.2 | 85.9 | 85.5 | 68.9 | 91.2 | 82.1 | 65.2 | 89.4 | 80.4 |
| CrossBERT 1B | 76.9 | 89.0 | 85.4 | 72.5 | 93.4 | 81.6 | 61.0 | 88.2 | 81.0 |
| BERT 2B* | 69.9 | 84.9 | 77.6 | 68.6 | 79.6 | 77.9 | 40.1 | 79.1 | 72.2 |

\* Training done under different hyperparameters

*Table 9.* Zoom on 1B scale model with 1e21 compute on Frozen Glue

## J. List of Models with hyperparameters

**BERT Model Configurations**

| ID | Params (M) | FLOPs | Dim | Layers | Heads | Head Dim | Steps | Tokens (B) | BSZ | GPUs | Grad Acc | LR | Min LR | $\beta_1$ | $\beta_2$ |
|---|---|---|---|---|---|---|---|---|---|---|---|---|---|---|---|
| 0 | 76.73 | 2.17e+18 | 576 | 12 | 9 | 64 | 15019 | 6.58 | 438272 | 4 | 1 | 2.75e-03 | 1.0e-6 | 0.90 | 0.95 |
| 1 | 103.28 | 4.64e+18 | 640 | 14 | 10 | 64 | 18268 | 9.63 | 527360 | 1 | 5 | 2.29e-03 | 1.0e-6 | 0.90 | 0.95 |
| 2 | 151.85 | 1.14e+19 | 768 | 16 | 12 | 64 | 24783 | 15.10 | 609280 | 1 | 7 | 1.85e-03 | 1.0e-6 | 0.90 | 0.95 |
| 3 | 226.70 | 2.83e+19 | 896 | 18 | 14 | 64 | 29287 | 23.78 | 812032 | 1 | 13 | 1.48e-03 | 1.0e-6 | 0.90 | 0.95 |
| 4 | 334.06 | 6.73e+19 | 1024 | 22 | 16 | 64 | 36291 | 36.68 | 1010688 | 3 | 7 | 1.21e-03 | 1.0e-6 | 0.90 | 0.95 |
| 5 | 503.81 | 1.65e+20 | 1152 | 28 | 9 | 128 | 46771 | 57.47 | 1228800 | 6 | 5 | 9.72e-04 | 1.0e-6 | 0.90 | 0.95 |
| 6 | 795.27 | 4.24e+20 | 1408 | 30 | 11 | 128 | 58575 | 92.13 | 1572864 | 16 | 3 | 7.75e-04 | 1.0e-6 | 0.90 | 0.95 |
| 7 | 1208.24 | 1.00e+21 | 1664 | 33 | 13 | 128 | 72072 | 141.70 | 1966080 | 40 | 3 | 6.30e-04 | 1.0e-6 | 0.90 | 0.95 |

*Table 10.* BERT model configurations across different scales. Parameters are in millions (M), Tokens in billions (B). BSZ = total batch size, Grad Acc = gradient accumulation steps.

**CrossBERT Model Configurations**

| ID | Params (M) | FLOPs | Enc Dim | Enc Layers | Enc Heads | Enc Head Dim | Pred Dim | Pred Layers | Pred Heads | Steps | Tokens (B) | BSZ | GPUs | Grad Acc | LR | Min LR | $\beta_1$ | $\beta_2$ |
|---|---|---|---|---|---|---|---|---|---|---|---|---|---|---|---|---|---|---|
| 0 | 61.80 | 1.92e+18 | 640 | 15 | 10 | 64 | 640 | 3 | 10 | 13745 | 6.19 | 450560 | 1 | 5 | 4.51e-03 | 1.0e-6 | 0.90 | 0.95 |
| 1 | 93.62 | 4.91e+18 | 768 | 17 | 12 | 64 | 768 | 4 | 12 | 19749 | 9.91 | 501760 | 1 | 7 | 3.47e-03 | 1.0e-6 | 0.90 | 0.95 |
| 2 | 138.58 | 1.15e+19 | 896 | 19 | 14 | 64 | 896 | 4 | 14 | 26978 | 15.19 | 563200 | 1 | 11 | 2.73e-03 | 1.0e-6 | 0.90 | 0.95 |
| 3 | 205.57 | 2.72e+19 | 1024 | 23 | 16 | 64 | 1024 | 5 | 16 | 39850 | 23.34 | 585728 | 1 | 13 | 2.15e-03 | 1.0e-6 | 0.90 | 0.95 |
| 4 | 315.61 | 6.83e+19 | 1152 | 29 | 9 | 128 | 1152 | 7 | 9 | 53691 | 36.95 | 688128 | 3 | 7 | 1.67e-03 | 1.0e-6 | 0.90 | 0.95 |
| 5 | 494.24 | 1.70e+20 | 1408 | 31 | 11 | 128 | 1408 | 7 | 11 | 79141 | 58.35 | 737280 | 6 | 5 | 1.29e-03 | 1.0e-6 | 0.90 | 0.95 |
| 6 | 757.47 | 4.07e+20 | 1664 | 34 | 13 | 128 | 1664 | 8 | 13 | 114660 | 90.17 | 786432 | 16 | 3 | 1.01e-03 | 1.0e-6 | 0.90 | 0.95 |
| 7 | 1176.29 | 9.99e+20 | 1920 | 41 | 15 | 128 | 1920 | 10 | 15 | 176106 | 141.38 | 802816 | 56 | 1 | 9.00e-04 | 1.0e-6 | 0.90 | 0.95 |
| 8 | 5411.7 | 5.0e21 | 2560 | 52 | 20 | 128 | 2560 | 13 | 20 | 304688 | 319.49 | 1048576 | 128 | 1 | 6.00e-04 | 1.0e-6 | 0.90 | 0.95 |

*Table 11.* CrossBERT model configurations across different scales. CrossBERT includes both an encoder and a predictor. Parameters are in millions (M), Tokens in billions (B). BSZ = total batch size, Grad Acc = gradient accumulation steps.

