# OpenReview forum: "Separating Representation from Reconstruction Enables Scalable Text Encoders"
_ICML.cc/2026/Conference — ICML 2026 regular_

### Official Review · Reviewer_Czg5 · 2026-03-08

**Soundness:** 4
**Presentation:** 3
**Significance:** 3
**Originality:** 3
**Overall Recommendation:** 5
**Confidence:** 4

**Summary:**

The paper introduces a new text embedding model called CrossBERT, involving a new co-designed architecture and training paradigm. The key motivation of the new techniques is to decouple direct token prediction and representation learning. The technique is called "Complementary Masking Strategy" (CMS) and consists of, for each incoming sequence, masking some % of the tokens, passing the unmasked part through an encoder; passing the masked part through a more lightweight "predictor" which does not use self-attention but only cross-attends to the features of the unmasked part at each layer; and using the features of the masked tokens to predict the token indices that were masked.  Experiments show that it performs well at linear probing and text embedding tasks relative to BERT with improved token efficiency.

**Compliance With Llm Reviewing Policy:**

Affirmed.

**Final Justification:**

The initial reading of the paper was positive, the rebuttal answered my questions/follow-ups satisfactorily, and so I continue to recommend acceptance.

**Key Questions For Authors:**

1. Can you show the training stability (preferably as the model scales), relative to baselines?
2. Can you clarify why the predictor doesn't learn to ignore the encoder and pass through the masked tokens' embeddings in order to recover exactly the prediction of the masked tokens? What pressure from the optimization method or the system stops this?
3. (Not required but just wondering --) Did you try to extend this approach to multimodal embedding? Are there any special caveats about the mask shape/assignment there?

**Limitations:**

yes

**Strengths And Weaknesses:**

**Strengths:**
- The empirical results are strong. Notably, the new model appears better than both the classic BERT and more modern alternatives such as ModernBERT on representation learning (measured by linear probing and MTEB). The scaling curves look promising as well. The data efficiency from CMS also is a nice improvement.
- The careful empirical analysis and study is thorough, even when not directly measuring performance versus baselines but rather checking various ablations and understanding properties of the system.
- The paper is well-presented and overall easy to read.

**Weaknesses (see questions):**
- There is not that much discussion of training stability and how it changes over scales. In self-supervised learning this kind of joint-embedding approach can often have unstable training.

---

> ### Author Rebuttal · Authors · 2026-03-31
>
> We thank the reviewer for their positive feedback and we appreciate their suggestion. We address their concerns below.
>
> Questions
> 1. In appendix D, we show the training loss at different scales, but we can include the following plot, [performance evolution during training](https://i.postimg.cc/DyqLNCFR/image.png), showing the evolution of downstream performance during different the training.
> 2. The predictor has no self-attention; it is simply a cross-attention block and MLPs. The input to the predictor are “information-empty”, \<MASK\> tokens. Therefore if the predictor ignored the encoder the objective would be unfeasible by construction. Unlike T5 or other encoder-decoder architectures where the decoder takes as input the shifted tokens and has self + cross-attention, the decoder can learn to ignore the encoder. There is no specific optimization method since the constraint is imposed structurally, but we can add further explanation in the main text to further emphasize/clarify this.
> 3. For the scope of this paper we have not tried, but indeed it is a good and interesting logical next step.

---

> > ### Author Rebuttal · Reviewer_Czg5 · 2026-04-03
> >
> > Thanks for the rebuttal. Overall my questions have been answered and I continue to recommend acceptance.
> >
> > > In appendix D, we show the training loss at different scales, but we can include the following plot, performance evolution during training, showing the evolution of downstream performance during different the training.
> >
> > Yes, this plot is great, you probably should put it in. It even looks more stable than BERT, though this is hard to conclusively prove without much more evidence.
> >
> > > The predictor has no self-attention; it is simply a cross-attention block and MLPs. The input to the predictor are “information-empty”, <MASK> tokens. Therefore if the predictor ignored the encoder the objective would be unfeasible by construction. Unlike T5 or other encoder-decoder architectures where the decoder takes as input the shifted tokens and has self + cross-attention, the decoder can learn to ignore the encoder. There is no specific optimization method since the constraint is imposed structurally, but we can add further explanation in the main text to further emphasize/clarify this.
> >
> > I understand now. You may want to change Figure 2 so the 1 4 6 tokens aren't sitting at the bottom of the vertical network (same as the other tokens sit at the bottom of the left network, since they are inputs) to avoid this confusion.

---

### Official Review · Reviewer_9kKz · 2026-03-13

**Soundness:** 2
**Presentation:** 3
**Significance:** 2
**Originality:** 2
**Overall Recommendation:** 3
**Confidence:** 4

**Summary:**

This paper proposes CrossBERT, a decoupled architecture designed to address the degradation of representation quality in encoders during scaling. The core idea is to physically isolate semantic learning from token reconstruction by introducing two distinct modules: an Encoder responsible for representation and a lightweight Predictor responsible for reconstruction.

**Compliance With Llm Reviewing Policy:**

Affirmed.

**Key Questions For Authors:**

1. Do the authors plan to validate this degradation trend and the effectiveness of CrossBERT at a larger parameter size (e.g., 7B or higher)?

2. In abstract, you claim "This suggests a misalignment between direct token prediction and the learning of rich, versatile, easily extractable representations," but why? Could you explain this further?

**Limitations:**

No. While the authors discuss the efficiency of the architecture, the following limitations should be more explicitly addressed to provide a balanced view:

The authors observe a degradation trend, but only up to 1.21B parameters. A significant limitation is the lack of verification at the 7B+ scale, where emergent properties in other architectures typically appear. This leaves the "universal" nature of the observed degradation in question.

**Strengths And Weaknesses:**

Strength:

This method offers a novel approach to encoder design. By physically isolating representation learning from label reconstruction, it challenges the monolithic architecture tradition that has existed since BERT, providing an innovative decoupled perspective for improving encoder scaling capability.

Weakness:

1. The authors attribute BERT’s performance degradation to the MLM objective but lack a rigorous theoretical explanation. A notable contradiction exists: Decoder-only models (e.g., the GPT series) also perform token prediction tasks, but the semantic understanding benefits significantly from scaling (the Scaling Law). Why does a similar prediction task facilitate semantic abstraction in Decoder architectures but act as a distractor in BERT? The paper fails to provide a fundamental explanation for this architectural disparity from the perspective of information theory or representation learning.

2. The core conclusions are based on observations of models with a maximum size of only 1.21B parameters. In the current era of Large Language Models, 1.21B is insufficient to represent true scaling trends. Could the observed degradation be a local phenomenon specific to this scale? If standard BERT were scaled to 10B or more, would it exhibit a leap in capabilities (Emergence) similar to that of Decoder models?

3. While Table 2 details CrossBERT’s stability across various masking ratios, the paper completely lacks a comparative experiment training standard BERT at high masking ratios (e.g., 50%). It remains unclear whether CrossBERT’s success stems from architectural decoupling or simply from the training gains from a more challenging task. If standard BERT also shows significant improvement in representation quality at a 50% masking ratio, the necessity of the Predictor module would be greatly diminished.

---

> ### Author Rebuttal · Authors · 2026-03-31
>
> We thank the reviewer for their positive feedback and constructive criticism. We address each of their concerns below.
>
> > The core conclusions are based on observations of models with a maximum size of only 1.21B parameters.
> > The authors observe a degradation trend, but only up to 1.21B parameters. A significant limitation is the lack of verification at the 7B+ scale, where emergent properties in other architectures typically appear. This leaves the "universal" nature of the observed degradation in question.
>
> During the training of our BERT 1.21B and 795M variants, we observed bad frozen representation (fig. 1). Given this trend, we initially bypassed the BERT baseline at 5e21 FLOPS (~2.4B parameters). However, to further document this degradation, we are launching a ~2.4B BERT training run and will update Figures 1, 3, and 4 accordingly in the final version.
>
> In appendix D we have shown the learning curves of BERT as it scales. We have evaluated them on a separate validation set not correlated to the training set. In Fig. 5 we can see that the scaling law is accurate, the validation losses decrease as expected.
>
> > Could the observed degradation be a local phenomenon specific to this scale? If standard BERT were scaled to 10B or more, would it exhibit a leap in capabilities (Emergence) similar to that of Decoder models?
>
> B) Following the above arguments, all experimental results for standard BERT suggest that scaling to 7B+ encoders will not lead to emergent behaviors. Of course this claim is hard to falsify because the lack of emergent properties at 1B or 5B does not prohibit their existence at 10B, 20B, 50B etc. Scaling encoders to 7B+ regimes would be the scope of an entirely different paper.
>
> Unlike decoder models, encoders have very good performance on text embedding tasks on small scales. Scaling laws for encoders are not the same as those of decoders. Encoders require much smaller models and much larger data to scale efficiently. Hence why to the best of our knowledge there are no SOTA pre-trained encoders-only bigger than ~3B scale.
>
> > While Table 2 details CrossBERT’s stability across various masking ratios, the paper completely lacks a comparative experiment training standard BERT at high masking ratios (e.g., 50%).
>
> C) We agree with the reviewer and have added the results training standard BERT to different masking ratios, [BERT Masking](https://rentry.co/p4we37nz) as it was done in preliminary study. In summary, the performance declines with higher masking ratios. These results further support our claim. While early BERT models defaulted to a 15% masking ratio, recent literature [1, 2] demonstrates that the optimal rate is between 20–40% and is now the standard masking ratio for recent BERT implementation.
>
> > The authors attribute BERT’s performance degradation to the MLM objective but lack a rigorous theoretical explanation.
>
> D) We have not been able to find a theoretical explanation detailing the failure of MLM. Our analysis is based on practical reasoning, experimental results such as Table 1 and the discussion L110-114. A rigorous theoretical explanation would strengthen our experimental conclusions; it would be an interesting avenue for future research. We can add a discussion of this limitation in the main text.
>
> Questions:
> 1. As mentioned in A) we will update to a ~2.4B scale, but due to practical compute considerations we may not be able to go higher than ~2.4B.
> 2. Our experiments on BERT return two seemingly contradictory results: the perplexity/validation loss are scaling properly but the downstream performance is getting worse. So as BERT scales its reconstruction is improving but its representations are degrading. This result highlights that while scaling laws for validation loss remain predictive, they do not inherently guarantee better features. Furthermore, as can be seen in Table 1, BERT downstream performance peaks for some intermediary activations, not the final layer. This suggests that the final layers of BERT are focused only on reconstructing the token in order to lower the MLM loss as much as possible.
> 3. The reviewer mentioned previously NTP and the seeming contradiction with the argument above. Note that even large decoders trained with NTP are not good text-embedders out-of-the-box. They have to go through extensive post-training pipelines to become good text-embedders (LLM2Vec, Qwen, Gemma). So it is not impossible that the same problem we have identified here for MLM with encoders also exists for decoders with NTP, it is simply masked by the amount of post-training done on the pre-trained backbone. However, to prove such claims for NTP would require a separate detailed study.
>
> We hope these revisions and clarifications fully address your concerns.
>
> [1] Wettig, A., et al. "Should you mask 15% in masked language modeling?." EACL. 2023.
> [2] Portes, J., et al. "MosaicBERT: A bidirectional encoder optimized for fast pretraining." NeurIPS. 2023.

---

### Official Review · Reviewer_f7SL · 2026-03-16

**Soundness:** 2
**Presentation:** 2
**Significance:** 3
**Originality:** 3
**Overall Recommendation:** 3
**Confidence:** 3

**Summary:**

The paper identifies a representation degradation phenomenon in standard Masked Language Models (MLM) like BERT when scaled up and evaluated under frozen protocols. To address this, the authors propose CrossBERT, a bipartite architecture that decouples representation learning from token reconstruction using a lightweight cross-attention predictor. By employing a Complementary Masking Strategy  and high masking ratios, the model achieves a 1.5-2x speedup in throughput and doubles sample efficiency. The authors provide a comprehensive scaling analysis from different FLOPs, demonstrating that CrossBERT maintains monotonic performance gains in frozen evaluations where standard BERT fails.

**Compliance With Llm Reviewing Policy:**

Affirmed.

**Key Questions For Authors:**

- The proposed architecture resembles masked autoencoder-style asymmetric reconstruction frameworks. What are the key conceptual differences between CrossBERT and existing asymmetric masked modeling approaches, and why should the proposed design lead to fundamentally better representations?
- The paper argues that MLM leads to suboptimal representations. Can the authors provide stronger empirical evidence isolating the effect of the architectural change from differences in training scale, training data, or optimization details?
- Can the authors provide additional ablation studies to clarify the contribution of the complementary masking strategy relative to the architectural changes?

**Limitations:**

yes

**Strengths And Weaknesses:**

## Strengths
- The discovery that standard MLM scaling leads to features that are increasingly unexploitable by frozen probes is significant and challenges the current reliance on full finetuning.
- The proposed architecture is relatively simple and easy to implement. The separation between the encoder and the reconstruction predictor is intuitive and could potentially improve training efficiency. The complementary masking strategy is also a practical technique that may increase token utilization during pretraining.
- The paper includes scaling experiments across multiple model sizes, compute budgets, and commonly used benchmarks.

## Weaknesses
- In the core scaling analysis (Figures 3 and 4), CrossBERT is only compared against a standard BERT baseline. It lacks comparisons with other efficient MLM variants like ELECTRA or DeBERTa, which also aim to improve sample efficiency and representation quality.\
- The core claim that MLM fundamentally leads to poor representations is not sufficiently supported. Modern encoder models trained with MLM variants (e.g., RoBERTa-style objectives and recent embedding models) already achieve strong representation quality. The paper does not convincingly isolate whether the improvements come from the architectural change itself or from differences in training scale and data.
- The experiments rely heavily on frozen encoder evaluation but omit comparisons with strong modern embedding methods and alternative pretraining objectives. Without these baselines, it is difficult to determine whether CrossBERT actually advances the state of the art in representation learning.

---

> ### Author Rebuttal · Authors · 2026-03-31
>
> We thank the reviewer for their positive feedback and constructive criticism. We address each of their concerns below in order to improve our paper.
>
> > In the core scaling analysis (Figures 3 and 4), CrossBERT is only compared against a standard BERT baseline. It lacks comparisons with other efficient MLM variants like ELECTRA or DeBERTa, which also aim to improve sample efficiency and representation quality.
>
> A) We focused mainly on the modern BERT baseline (MLM with 20% masking ratio, no next-sentence-prediction, RMSNorm, no bias, RoPE, training with optimal data-to-model ratio) since it is the current SOTA (ModernBERT, NeoBERT) largely outperforming DeBERTaV3 and RoBERTa [1].
>
> DeBERTaV3 contains architecture changes which are incompatible with modern techniques such as RoPE, making it harder to draw a fair comparison to our models. However we agree that ELECTRA, which replaces the MLM objective with RTD (replaced token detection), is an interesting comparison to make since it can easily be modernized. Following the reviewers suggestions we have run experiments for ELECTRA and updated our results/conclusions accordingly: [Electra Benchmarks](https://rentry.co/hhbm8t4c).
>
> These results show that the modernized ELECTRA has a strong performance on GLUE tasks but performs terribly on MTEB, significantly worse than our BERT baseline, especially in retrieval. Therefore replacing MLM by RTD improved GLUE performance but led to other unsolved issues which are not the focus of our paper [2]. The conclusion is coherent with the current literature [1],[2] where ELECTRA style models like DeBERTa V3 yield bad representations.
>
> The advantage of CrossBERT is a better understanding of the “failure-mode” of MLM and the targeted fix addressing it.
>
> > Modern encoder models trained with MLM variants (e.g., RoBERTa-style objectives and recent embedding models) already achieve strong representation quality. The paper does not convincingly isolate whether the improvements come from the architectural change itself or from differences in training scale and data.
> > Without these baselines, it is difficult to determine whether CrossBERT actually advances the state of the art in representation learning.
>
> B) Given the reviewers comments we realize that we have not sufficiently emphasized that our BERT baseline is a modernized model following RoBERTa’s objective, inspired by NeoBERT which uses the latest Llama-like techniques. The only difference between the BERT baseline and CrossBERT is that our proposed architecture changes from “flat” to bipartite. All of our baselines have been run with the same training scale and data.
>
> We do not compare with recent embedding models like Qwen3 or Gemma because they have different datasets, mid-training or post-training recipes, whereas CrossBERT has a simple pretraining on a single dataset.  Our paper aims to improve pretraining embedding quality and reduce the gap between pretrained encoders and heavily post-trained encoders.
>
> We never claim CrossBERT is SOTA on representation quality. We simply show that among models that are only pre-trained and do not rely on heavy specific fine-tuning, CrossBERT has an advantage. But we agree with the reviewer that an interesting avenue of future research would be to implement a heavy post-training pipeline on CrossBERT to match/beat the SOTA.
>
> > The core claim that MLM fundamentally leads to poor representations is not sufficiently supported.
>
> C) We do not claim that MLM fundamentally leads to poor representations, in fact CrossBERT uses MLM. However we show experimentally that MLM on the flat architecture of BERT has a critical flaw (Table 1) which leads to poor representations when scaling. CrossBERT solves this flaw and therefore can scale.
>
> Questions
> 1. As stated in L117, CrossBERT was inspired by MAE in vision, but to the best of our knowledge no such model exists for text. In a certain sense CrossBERT is the first text asymmetric MAE.  On top of the architectural changes, we introduce CMS and we do not throw away the predictor but use it to warm-start the fine tuning process. (L160, Appendix C).
> 2. We believe to have answered the concerns in A),B) and C). Perhaps something that was not clearly stated in the paper is that our BERT baseline is not the classical BERT from Devlin et al. but the modernized architecture whose hyperparameters and changes are detailed in Appendix F. We will emphasize this more clearly in the main text.
> 3. We have provided a CMS ablation study in Appendix B. Could the reviewer specify what additional experiments they might require?
>
> We hope the proposed changes effectively address your feedback and clarify our contributions.
>
> [1] Warner, B. et al. (2025). Smarter, better, faster, longer: A modern bidirectional encoder. In Proc. ACL.
> [2] Rep et al. (2024). Are ELECTRA’s sentence embeddings beyond repair? Findings of EMNLP.

---

> > ### Author Rebuttal · Reviewer_f7SL · 2026-04-03
> >
> > Thank you for your reply to my concerns. But as it stands now, it may need significant revision. So i stick to my score, but I do not object to the acceptance of this paper.

---

### Decision · Program_Chairs · 2026-04-30

**Decision:**

Accept (regular)

**Comment:**

This paper proposes an approach for training language representations through encoder-only MLMs. The approach involves an MAE-like approach where an encoder processes the unmasked tokens, and the encoder's representation is cross-attended to by a separate predictor that just conditions on the masked tokens. This approach is found to enable the training of much more data-efficient encoders than ModernBERT, and also result in representations whose linear probe performance increases as MLM decreases (unlike in traditional MLMs).

While the idea is simple, it is novel to my knowledge, and convincingly backed up by empirical experiments. There were some concerns with regard to comparison against baselines such as ELECTRA, but these were largely addressed in the rebuttal in my opinion. I think this is a solid paper.

As noted by the authors, this type of encoder/decoder bipartite structure training was also adopted by T5. While I agree with the authors that T5 was primarily targeting the pretraining of seq2seq models instead of representation learning models, it might be useful to also compare CrossBERT against T5-style pretraining for representation learning purposes (which roughly corresponds to adopting an autoregressive structure on the decoder side).